# Updating age-specific contact structures to match evolving demography in a dynamic mathematical model of tuberculosis vaccination

Chathika Krishan Weerasuriya[1]*, Rebecca Claire Harris[1¤a], Christopher Finn McQuaid[1], Gabriela B. Gomez[2¤b], Richard G. White[1]

**1** TB Modelling Group, TB Centre and Centre for the Mathematical Modelling of Infectious Diseases, Department of Infectious Disease Epidemiology, Faculty of Epidemiology & Population Health, London School of Hygiene and Tropical Medicine, London, United Kingdom, **2** Department of Global Health & Development, Faculty of Public Health & Policy, London School of Hygiene and Tropical Medicine, London, United Kingdom

¤a Current address: COVID-19 Medical Franchise, Sanofi Pasteur, Singapore, Singapore
¤b Current address: Department of Modelling, Epidemiology and Data Sciences, Sanofi Pasteur, Lyon, France
* c.weerasuriya@lshtm.ac.uk

**Data Availability Statement:** Model datasets and code are available on GitHub: https://github.com/weerasuriya/tbconvax-pcb. A snapshot is archived

## Abstract

We investigated the effects of updating age-specific social contact matrices to match evolving demography on vaccine impact estimates. We used a dynamic transmission model of tuberculosis in India as a case study. We modelled four incremental methods to update contact matrices over time, where each method incorporated its predecessor: fixed contact matrix (M0), preserved contact reciprocity (M1), preserved contact assortativity (M2), and preserved average contacts per individual (M3). We updated the contact matrices of a deterministic compartmental model of tuberculosis transmission, calibrated to epidemiologic data between 2000 and 2019 derived from India. We additionally calibrated the M0, M2, and M3 models to the 2050 TB incidence rate projected by the calibrated M1 model. We stratified age into three groups, children (<15y), adults (≥15y, <65y), and the elderly (≥65y), using World Population Prospects demographic data, between which we applied POLYMOD-derived social contact matrices. We simulated an M72-AS01$_E$-like tuberculosis vaccine delivered from 2027 and estimated the per cent TB incidence rate reduction (IRR) in 2050 under each update method. We found that vaccine impact estimates in all age groups remained relatively stable between the M0–M3 models, irrespective of vaccine-targeting by age group. The maximum difference in impact, observed following adult-targeted vaccination, was 7% in the elderly, in whom we observed IRRs of 19% (uncertainty range 13–32), 20% (UR 13–31), 22% (UR 14–37), and 26% (UR 18–38) following M0, M1, M2 and M3 updates, respectively. We found that model-based TB vaccine impact estimates were relatively insensitive to demography-matched contact matrix updates in an India-like demographic and epidemiologic scenario. Current model-based TB vaccine impact estimates may be reasonably robust to the lack of contact matrix updates, but further research is needed to confirm and generalise this finding.

and available via Zenodo: https://doi.org/10.5281/zenodo.5902656.

**Funding:** C.K.W. is funded by UKRI/MRC (MR/N013638/1). R.G.W. is funded by the Welcome Trust (218261/Z/19/Z), NIH (1R01AI147321- 01), EDTCP (RIA208D-2505B), UK MRC (CCF17-7779 via SET Bloomsbury), ESRC (ES/P008011/1), BMGF (OPP1084276, OPP1135288 and INV-001754), and the WHO (2020/985800-0). C.F.M is funded by the Bill and Melinda Gates Foundation (TB MAC OPP1135288). The funders had no role in study design, data collection and analysis, decision to publish, or preparation of the manuscript.

**Competing interests:** I have read the journal's policy and the authors of this manuscript have the following competing interests: G.B.G. and R.C.H. report currently being employed at Sanofi Pasteur, unrelated to this work or TB.

## Author summary

Mathematical models are increasingly used to predict the impact of new and existing tools, e.g., vaccines, that aim to control the transmission of infectious diseases. Within these models, investigators often assume that individuals contact each other according to specific patterns, particularly between and within different age groups. These patterns are typically derived from surveys of social contact or other models and reflect the particular age composition of their source population. However, when models are set over long time scales, e.g., decades, population age composition is likely to change. Despite this reality, few models update their contact patterns to match changing age composition. Furthermore, none have assessed whether their final estimates of disease-control intervention impact are affected by updating contact patterns. We measured whether different techniques to update social contact patterns to match evolving demography produce different vaccine impact estimates, using a mathematical model of tuberculosis set in an India-like scenario between 2025–2050. We found that vaccine impact was stable across a range of different update methods. Thus, existing model-based vaccine impact estimates may be stable to a lack of these updates, but further work is required to confirm these findings.

## Introduction

Social contact patterns are a crucial contributor to the patterns observed in communicable disease epidemiology [1,2]. Contacts can be grouped by various criteria, including age groups or gender, behavioural characteristics (e.g., high- or low-risk behaviours), or location (e.g., at home, school, or workplace) [3]. Among these, age-specific mixing is a key contributor to the behavioural drivers of age-specific burden in many infectious diseases (e.g., measles, mumps, or tuberculosis). To capture these contributions, dynamic infectious disease models increasingly implement age-specific contact matrices [1,4]. These matrices present the number of contacts that each member of some age group *i* makes with members of each age group in the model over a defined time period (typically daily). Multiple recent studies have attempted to characterise such age-specific contact patterns. For example, the POLYMOD study by Mossong et al. [5] provides comprehensive empirical nationally-representative estimates of age-dependent contact rates (expressed as contact matrices) in eight European countries based upon contact diaries. Prem et al. [3] have estimated synthetic contact matrices for a wide range of countries using results from POLYMOD, Demographic and Health Surveys, demographic data, and other sources. Estimates of subnational or localised contact rates and mixing patterns have also been published for China [6], India [7], Zimbabwe [8] and Kenya [9], among others. Most often, contact surveys request that participants record every contact they have with others within a fixed time period in a diary, where a contact is defined as at least a skin-to-skin (e.g., a handshake) or a two-way conversation. Participants are typically recruited to reflect the sociodemographic characteristics of their source populations.

The improved representation of inter-age transmission dynamics conferred by integrating a heterogeneous age-specific contact structure is particularly important in modelling studies which investigate age-targeted interventions (e.g., vaccines). As such, the specific implementation of age-specific mixing is a key structural choice made during model design. Modellers must consider whether the contact structure will interact with changes to other model features, chiefly, demography.

Contact matrices reflect a snapshot of contact patterns at a particular time. The time point corresponds to contact survey dates for empirical estimates or some appropriate mid-point for

data included in synthetic matrices. Each contact matrix is co-determined by the intrinsic preferences of groups for contact with other groups ("assortativity") and the demographic composition of its source population. Fundamentally, a contact matrix in a population with a given age structure should demonstrate 'reciprocity of contacts', where the total number of contacts between all members of some age group $i$ with another age group $j$ equals the total number of contacts between all of group $j$ with group $i$. If total contacts between age groups are unbalanced, this leads to an error in the calculated age-specific force of infection parameters within the model. This propagates into errors in age-specific burden and introduces further error into the estimated impact of age-targeted interventions.

In infectious disease models with a short time-horizon, we can reasonably assume that the contribution of demographic composition to age-specific contact patterns remains relatively constant. However, this assumption is violated when the demographic structure is expected to change, as is the case when modelling diseases with long latency periods (e.g., tuberculosis and syphilis) or interventions where impact is expected to be long term (e.g., vaccines). In these circumstances, matrices must be updated to ensure contact reciprocity.

However, corrections to ensure reciprocity may alter underlying properties of the matrix, including assortativity (the relative preference of one group for contact with another) and the overall population-wide average contact rate. Depending on the specific research question, modellers may wish to preserve some or all these properties. For example, current evidence suggests that targeting TB vaccines to the elderly (in whom TB burden is concentrated) in China is most likely to be impactful as its population continues to age. Under these circumstances, failure to preserve contact assortativity may bias the estimates of elderly targeted vaccination.

Arregui et al. [10] describe methods to project a contact matrix estimated from any particular population to a population with an arbitrarily different demographic structure while variably preserving reciprocity, assortativity, and overall average contact rate. Despite this development, we are aware of only one study [1] investigating whether such contact matrix updates affect model-based predictions of disease burden. This study of *Mycobacterium tuberculosis* transmission suggested that a lack of demographically matched contact matrix updates might underestimate future TB burden. In addition, no studies have investigated if contact matrix updates affect dynamic model-based impact estimates infectious disease control interventions.

In this study, we hypothesised that changing age-dependent contact rates through different contact matrix update methods in an evolving demographic context would lead to differential transmission dynamics between age groups in a disease with age-specific burden patterns. Furthermore, these differential transmission dynamics would propagate through the direct and indirect (transmission dependent) effects of vaccination, leading to differential vaccine impact estimates.

We investigated the effects of updating age-specific social contact matrices to match evolving demography on dynamic transmission model-based vaccine impact estimates, using tuberculosis in India as a case study.

## Methods

### Transmission model

We developed a six-compartment dynamic model of *Mycobacterium tuberculosis* (M. tb) transmission in the R [11] and Julia [12] programming languages, building on previous studies [13–17]. A full description of model structure, parameterisation and calibration are provided in S1 Text.

In brief, the model represented six states: (1) naive to and susceptible to infection; (2) latently infected; (3) active infectious TB disease (bacteriologically positive); (4) non-infectious active TB disease (bacteriologically negative); (5) TB disease on-treatment and (6) recovered from disease through successful treatment or natural cure. In addition, we stratified all states by vaccination status.

Flows between states represented changes in TB natural history state or treatment status. Natural history flows included infection by M. tb followed either 'fast progression' to active disease or 'slow progression' to latent infection, conversion from non-infectious active disease to infectious active disease, natural cure from active disease to the recovered state, reactivation from latency, and relapse from recovered. Treatment-related flows included detection and initiation on treatment, treatment success and recovery, and treatment failure leading to re-entry into non-infectious active disease.

We modelled ages 0–99, stratified into children (<15y), adults (15–64y) and elderly (≥65y). Annual historical and projected future birth rates and all-cause mortality rates were obtained from the United Nations World Population Prospects 2019 India country profile [18]. New births entered the children group in the first time step of each year. Age-group specific annual all-cause mortality, adjusted to remove TB mortality (section A.1 in S1 Text), was applied at the beginning of each time step.

We ran the model over 1950–2050 using a six-month timestep, calibrated to historical epidemiologic data over 2000–2019 and projected over 2020–2050.

We obtained prior ranges for natural history parameters from the literature, applying age-group specific ranges where possible. Rates of fast progression, reactivation from latency and TB mortality were constrained to be greater in children than adults. Rates of relapse from the recovered state and fast progression were constrained to be greater in the elderly than adults. Conversely, we constrained the natural cure rate and proportion of fast-progressors entering non-infectious disease to be lower in the elderly than adults.

## Social contact matrices

Empirical social contact data from India is limited to a study from one rural setting in Haryana [7]. These data were not nationally representative, and raw contact data were not published at the time of writing. As such, we used the SOCIALMIXR R package [19] to generate a base social contact matrix derived from the POLYMOD study [5], which aggregated empirical social contact survey data across 7,290 respondents and 97,904 contacts across eight European countries. Data from POLYMOD are expressed as the average number of contacts made per day by each survey participant per country, with participants binned into 5-year age groups. We used this data and the population size of the participant age group to generate the total contacts made by all participants in that age group. This was then aggregated into the specific groups in our model (0–14, 15–64, and 65+). Finally, totals were summed across all POLYMOD countries, then divided by total population to generate POLYMOD-wide average daily contact rates to generate a three age-group base matrix (Fig 1A) that corresponded to a snapshot of social contact at the time of the survey (2005–2006). This matrix reflected a source population that comprised approximately 16% children, 67% adults, and 17% elderly.

Arregui et al. [10] describe four methods to update contact matrices to match evolving demography, labelled M0–M3. We briefly describe the properties of each method below and in Table 1; detailed calculations and formulae are presented in the section B.2 in S1 Text. We adopted the same naming convention, referring to each independently calibrated model by its respective update method.

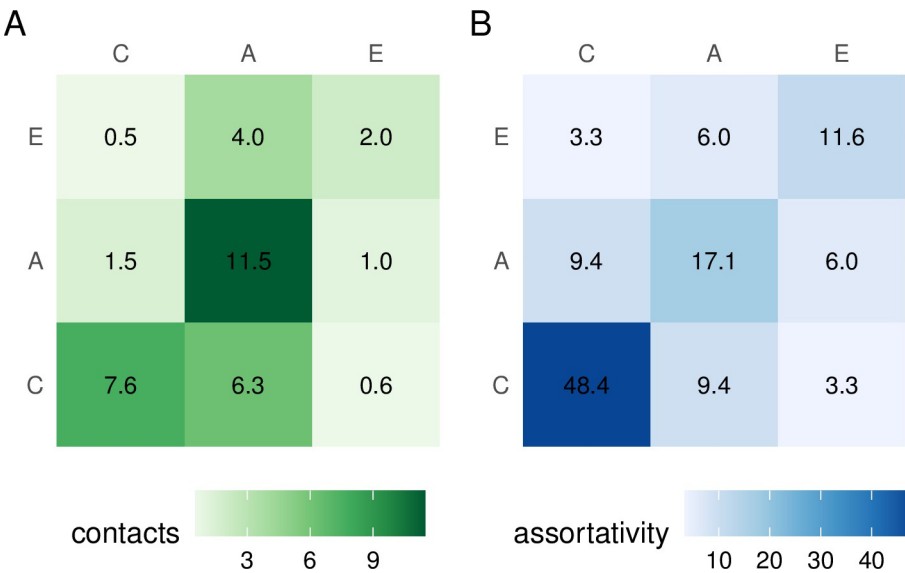

**Fig 1. Base contact and assortativity matrices.** A: contact matrix derived from the POLYMOD study, based on surveys conducted in 2005–2006, used without further transformation in the M0 model and with reciprocity correction in the M1 model. B: assortativity matrix derived from POLYMOD matrix, by decoupling the POLYMOD demographic structure from base contact matrix. Numbers within cells represent contact rates between the column-row age-group pairs. A = adults; C = children; E = elderly.

M0 represented the identity transformation, where contact rates remained invariant with changing demographics. The M1–M3 updates were incremental, such that M3 included the properties of M2, which included the properties of M1.

The M1 update method preserved *reciprocity*. Contact rates were adjusted to ensure that the total number of contacts between any age group $i$ and group $j$ was equal to total contacts between $j$ with $i$ as group sizes changed over time.

The M2 update method preserved *assortativity* in addition to reciprocity. Assortativity is the relative preference of some age group $i$ for contact with another group $j$, over that expected by homogeneous mixing between $i$ and $j$. For every contact matrix $Q$ with $n$ age classes, we can compute a corresponding assortativity matrix $R$. The entries of $R$ are the contact rates expected in a population where the $n$ classes are equally sized and where the relative preferences for contact between groups are the same as in $Q$, multiplied by $n$. We derived such an "assortativity matrix" from the M0 matrix (Fig 1B) by decoupling it from the original demographic structure of the POLYMOD survey. We then regenerated a contact matrix during each step of model run time by applying the demographic structure of that step to the assortativity matrix.

The M3 update method preserved the *average contact rate* in addition to reciprocity and assortativity. The M2 matrix was first used to calculate the average population-wide daily contact rate per individual and then normalised by this value.

**Table 1. Contact matrix transformation methods.** The properties preserved by each transformation M0–M3 are indicated as Yes (Y)/No (N) in the table. Adapted from Arregui et al [10].

| Property | M0 | M1 | M2 | M3 |
|---|---|---|---|---|
| Reciprocity | N | Y | Y | Y |
| Assortativity | N | N | Y | Y |
| Mean contact rate | N | N | N | Y |

Each of the methods described above implies different consequences on contact patterns due to demographic change. For example, as total contact volume remains constant with population size in M1, an ageing population would lead to more contacts between each child and older adults. In M2, the overall average contact rate may increase (or decrease) depending on the assortativity pattern and changes in the size of specific age classes, implying that more (or less) contact occurs between members in general. Finally, M3 implies the opposite: the total volume of contacts would grow in proportion to the total population. Which of the aforementioned update methods best reflects the true change in contact patterns is not known empirically; however, in this study, we use M1 as the intuitively "natural" base case against which other methods were compared.

We present the scaled effective contact rate ($\beta$)—where an effective contact was defined as sufficient to lead to infection, were it to occur between a susceptible and an infectious individual [20]—between each age-group pair to demonstrate evolving contact over time. In each contact matrix update scenario, we sampled an independent scaled probability of transmission per infectious contact ($\pi$), which we transformed, multiplied into the contact rate and scaled to a six-month timestep to compute $\beta$. Therefore, in all calibrated models,

$$\beta = -180\kappa \cdot \ln(1 - \pi),$$

where kappa ($\kappa$) represented the contact rate. Because each transformation scenario was calibrated independently (see below), $\beta$ parameters were difficult to compare across scenarios; we examine differences in $\beta$ parameter trends rather than magnitude.

## Country adapted matrix

In July 2021, Prem et al [21] published an updated set of synthetic contact matrices for 177 countries derived by adapting POLYMOD data to reflect country-level demographic, household, and location-specific composition (home, work, school, other) characteristics using UN World Population Prospects demographic projections, Demographic and Health Survey data, and other sources. We used the estimated number of total contacts predicted for India by Prem et al to generate an equivalent matrix to Fig 1A. We compared this back-calculated matrix to that in Fig 1A to assess if country-specific adjustments led to substantially different contact rates or assortativity at this level of age-group aggregation.

## Calibration

We calibrated four transmission models, labelled M0–3, using each of the contact matrix update methods. Each baseline (unvaccinated) scenario was fitted to overall rates of TB prevalence in 2015 [22], incidence in 2010 and 2019 [23,24], notifications in 2019 [23,24], and mortality in 2019 [23,24]. Age-specific incidence rates were not published at the time of writing. Therefore, we estimated incidence rates for <15, 15–99, and 65–99 year age groups using raw incidence estimates from WHO [23,24] and population estimates from World Population Prospects [18].

We captured historical programmatic control of TB by fitting treatment initiation rate to notification rate data, with treatment outcome rates per the WHO TB database [23].

We assumed M1 to be the "natural" base case against which to measure the other update methods. Further, to ensure that the baseline (unvaccinated) TB burden projected using all update methods remained comparable, allowing any differences in vaccine impact to be attributed to differential contact matrix updates, we calibrated models M0, M2, and M3 to the 2050 incidence rate projected by the fully calibrated M1 model.

Model calibration was performed in two stages. First, we used box-constrained optimisation to find initial parameter sets that fit all calibration targets. Second, we used these initial parameter sets to initialise an Approximate Bayesian Computation Markov chain Monte-Carlo (ABC-MCMC) sampler to fully characterise the parameter space compatible with the uncertainty ranges of the calibration targets. We extracted a final subsample of 1000 parameter sets from the ABC-MCMC chains for each contact matrix update method, with which the model was run to project baseline TB burden. We present median values as a measure of central tendency and minimum and maximum trajectories as uncertainty ranges. Posterior distributions for parameters and other calibration results are presented in sections D and E in S1 Text.

## Vaccine implementation

We simulated a 50% efficacy prevention of disease vaccine, effective in individuals with a prior history of TB infection (post-infection; PSI) that conferred 10-years of protection. Vaccination was delivered to populations without active disease and who were not receiving treatment.

We simulated vaccine delivery targeted to children, adults, or the elderly via 10-yearly mass campaigns that began in 2027 and achieved 70% coverage. Vaccine protection was modelled through a reduction (proportional to vaccine efficacy) in the rates of fast progression, reactivation from latency, and relapse from the recovered state. Vaccine waning was modelled as instantaneous at the end of protection. Details of the vaccine implementation are given in section B.5 in S1 Text.

We measured vaccine impact as the per cent incidence rate reduction in 2050 in vaccinated model runs compared to no-new-vaccine baseline runs.

We conducted sensitivity analysis by varying the host-infection status required for vaccine efficacy to include vaccines effective in individuals with no prior history of infection (pre-infection; PRI) and vaccines effective in individuals irrespective of TB infection history (pre- and post-infection; P&PI).

## Results

### Calibration and baseline trajectory

We calibrated to TB prevalence, incidence, notification, and mortality rates. The M1 model projected an overall 2015 prevalence rate of 217 (Uncertainty range (UR): 195–312) per 100,000, and incidence, mortality, and notification rates of 244 (UR: 205–265) per 100,000, 32 (UR: 30–35) per 100,000, and 167 (UR: 152–213) per 100,000, respectively, in 2019. The M1 model also projected an overall incidence of 234 (UR: 190–271) per 100,000 in 2050.

Overall TB incidence rates in the M0, M2, and M3 models were projected at 244 (UR: 206–265) per 100,000, 242 (UR: 209–266) per 100,000, and 225 (UR: 206–263) per 100,000, respectively in 2019. As expected, the projected incidence in 2050 for M0, M2, and M3 models remained within the envelope of the M1 projection. Projected 2050 incidence rates in M0 and M2, at 239 (UR: 195–271) per 100,000 and 258 (UR: 213–271) per 100,000, respectively, remained relatively stable compared to 2019. The 2050 projected median incidence rate in M3 rose slightly, with a narrowed uncertainty interval to 266 (UR: 237–271) per 100,000.

Age-specific incidence calibration for M1 and full calibration results for M0, M2, and M3 are presented in section D.3 in S1 Text. In general, we found a substantially lower TB burden in children than adults or the elderly (section D.3 in S1 Text). TB burden was comparable between adults and elderly (section D.3 in S1 Text). In addition, we found similar proportions of incident TB due to relapse, reactivation, or new infection followed by transmission across M0–M3 (section D.3.2 in S1 Text).

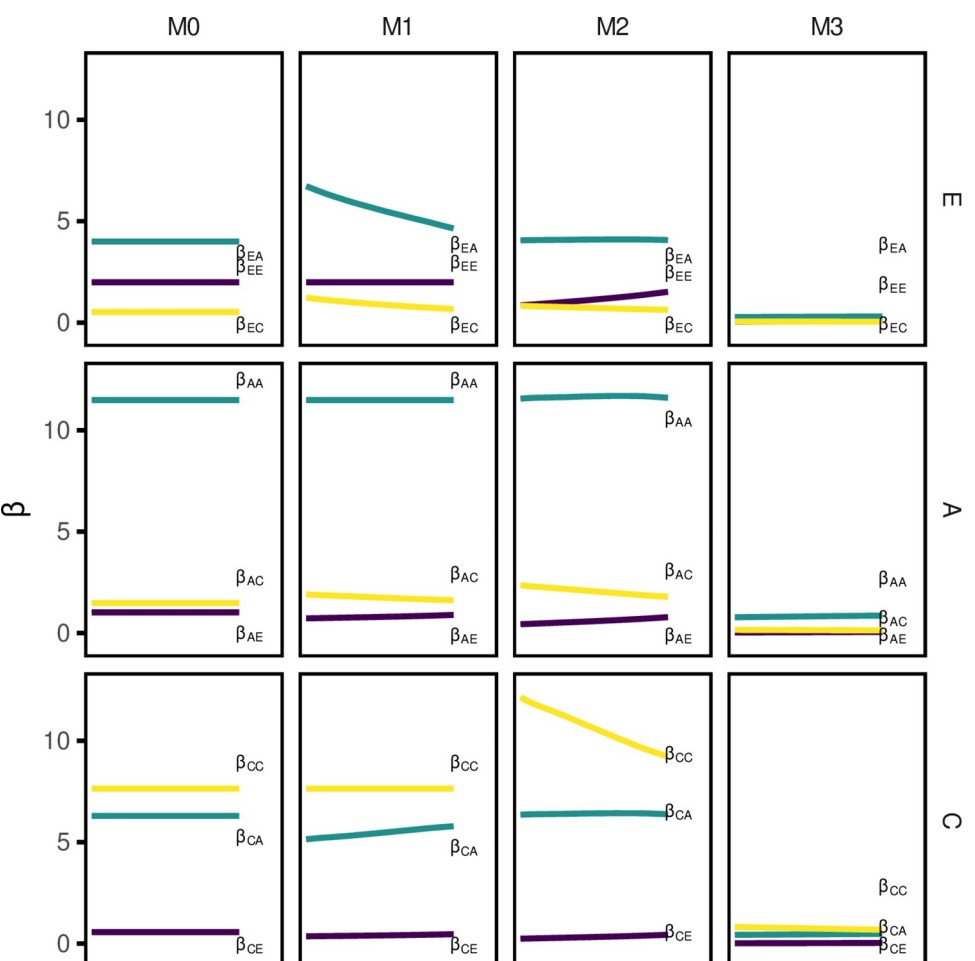

**Fig 2. Beta parameter change with time between all age group pairs.** Beta represents the effective contact rate between two age groups (subscripts) per six-month time step. Horizontal panels represent the age group making contacts; lines represent the beta parameter values corresponding to the number of contacts *with* the contactee group. A = adult, C = children, E = elderly.

## Evolution of contacts

The temporal evolution of the median per capita effective contact rates (β) for all age-group pairs are presented in Fig 2. The subscripts indicate the age classes of the individual and their contactee (A = adults; C = children; E = elderly). In addition, measures of reciprocity error, assortativity, and average contact rate differences between the update methods are presented in section B.2 in S1 Text.

In the M0 scenario, as expected, β values for all age-group pairs remain constant over time.

In the M1 model, the reciprocity correction ensured that within age-group β values remained constant over time and demographic change. Broadly, $\beta_{AC}$, $\beta_{AE}$, and $\beta_{EC}$ also remained relatively constant. $\beta_{EA}$ demonstrated the most marked decline, reflecting the distribution of a fixed total volume of contacts over a growing proportion of elderly. The opposite effect, albeit less marked, was seen in $\beta_{CA}$,

In the M2 model, values for $\beta_{AA}$, $\beta_{CA}$, $\beta_{CC}$, and $\beta_{EA}$ were higher than in the M0 or M1 models, reflecting the higher proportion of adults and children in India than in POLYMOD countries. Accordingly, values of $\beta_{CC}$ and $\beta_{AC}$ declined over 2025–2050, mirroring the declining

proportion of children in the population, suggesting fewer contacts between children and between each adult with children. We found the opposite effect in $\beta_{EE}$, $\beta_{AE}$, and $\beta_{CE}$: as the proportion of elderly in the population rose, the number of contacts with the elderly also rose.

β values and trends were similar between the M2 and M3 models. However, as the average contact rate declined in the M2 model over time (section B.2 in S1 Text) we found increasing trends in counterpart β values projected by the M3 model.

Finally, we found that adjustment for India-specific sociodemographic features did not lead to a substantially different base contact matrix compared to Fig 1A (Section F in S1 Text) at this level of aggregation of age-groups.

## Vaccine impact

A summary of vaccine impact, for a vaccine with 50% efficacy, conferring 10-years of protection, effective in individuals with a previous history of disease, and which prevented disease but not infection, is presented in Fig 3.

We found that vaccine impact estimates in all age groups remained relatively stable between the M0–M3 models, irrespective of vaccine targeting by age group. The maximum difference in impact, observed following adult-targeted vaccination, was 7% in the elderly, in whom we observed IRRs of 19% (uncertainty range 13–32), 20% (UR 13–31), 22% (UR 14–37), and 26% (UR 18–38) following M0, M1, M2 and M3 updates, respectively.

When the vaccine was delivered to adults, we observed an increasing vaccine impact in M0 through M3 models in all age groups. A similar across-model trend was seen in vaccine impact when vaccinating children, albeit of a smaller magnitude. A decreasing trend in vaccine impact from M0 to M3 was found in all age groups when vaccinating the elderly. However, we found substantial overlap in the uncertainty ranges of vaccine impact estimates across M0–M3 for all vaccine targeting and outcome combinations.

Overall findings were robust to variation in host-infection status required for efficacy; we found similarly stable vaccine impacts between M0–M3 for pre-infection or pre-and post-infection vaccines (section E in S1 Text).

## Discussion

We found that model-based estimates of TB vaccine impact in India remained stable over a range of simulated changes that matched contact structures to evolving demography.

Vaccine impact estimates in all age groups remained relatively stable between contact matrix update methods, irrespective of vaccine targeting by age group. The maximum difference in incidence rate reduction in 2050, observed following adult-targeted vaccination, was 7% in the elderly, in whom we observed IRRs of 19% (uncertainty range 13–32), 20% (UR 13–31), 22% (UR 14–37), and 26% (UR 18–38) following M0, M1, M2, and M3 updates, respectively.

Adult-targeted PSI vaccination led to the greatest vaccine impact in all age groups. In contrast, child- or elderly-targeted vaccination reduced TB burden within those groups with minimal indirect impact on others. This pattern suggests relatively low transmission of infection from children or the elderly to outside their age groups in the modelled epidemic.

The effective contact rate between children ($\beta_{CC}$) declined markedly over time in the M2 and M3 models, while in M0 and M1 it remained relatively constant. We found that targeting vaccines to children did not yield different vaccine impacts between M0–M3 despite this difference. This is likely because the burden of TB in children was very low in all models (section D.3 in S1 Text), leading to a correspondingly low force of infection originating from this group. Thus, the disease and transmission avertible by targeting vaccination to children was

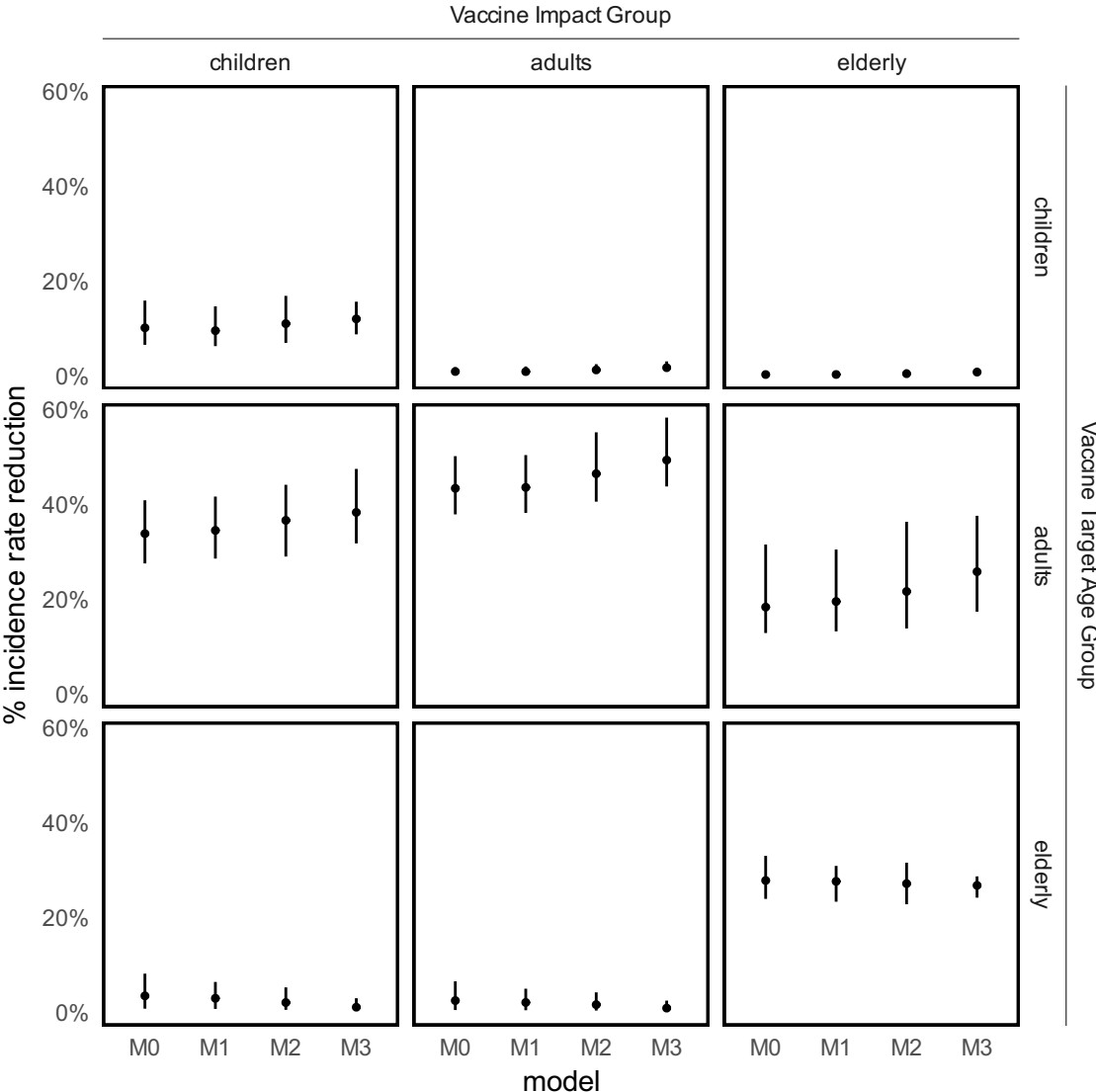

**Fig 3. Per cent TB incidence rate reduction in 2050 compared to no-vaccine baseline.** Rows indicate age group to which vaccine was targeted. Columns indicate in which age group impact was measured. Points indicate median estimate, bars indicate uncertainty range.

limited in all models, minimising differential impact. We note that, although TB vaccine impact is unchanged, differential contact matrix updates may substantially affect models of other more prevalent or more infectious childhood diseases.

In contrast to children, we found that disease burden in the elderly was greater than or comparable to adults. However, the elderly had lower contact rates with all age groups, with low and stable $\beta_{E*}$ and $\beta_{*E}$ values across M0–M3. This may also have contributed to lower avertible disease and transmission levels, reducing differential impact across the update methods.

Our findings may reflect the dominant contribution of intrinsic biological factors (represented in the model as constrained parameterisation and age-specific parameter values) over behavioural factors (i.e., age-specific contact patterns) to age-specific disease burden in TB. This would reduce the sensitivity of the force of infection to changes in contact rates,

contributing to the stability of vaccine impacts across M0–M3. However, this balance may differ in other diseases, warranting investigation on a per disease basis.

Finally, numerically, it can be shown that only the M2 and M3 transformations satisfy the frequency dependence assumption commonly used when calculating force of infection parameters in human dynamic transmission models. However, even in the M2 and M3 models, frequency dependence is only maintained if the population grows while maintaining a constant age composition, which is unlikely.

Our findings must be interpreted considering several limitations of this study.

Firstly, we used large age strata. Although granular contact data from the POLYMOD study were available, we were limited to stratifying the model into three relatively broad age groups by the resolution of available calibration data in India. As a result, subtle interactions of contact rates with evolving demography may have been obscured, especially within the wide adult age group. It is difficult to estimate the direction of bias this limitation might impose. However, most previous TB vaccine models have considered these age groups in aggregate, as we have done, as they are of interest from an epidemiologic and vaccine implementation strategy perspective [13–15].

Secondly, our case study of India was limited by the scarcity of granular, nationally representative TB epidemiologic data. India has not yet published a national survey of TB prevalence; nationally representative empirical estimates of age-specific prevalence are unavailable. In general, TB notification data are known to have age-specific biases, which may underestimate the burden of disease in children and older people [25,26]. Additionally, our use of a generic POLYMOD contact matrix rather than an India-specific matrix may reduce the accuracy of our findings. We know of two previous studies which estimated social contact patterns in India. Prem et al. [3] combined Demographic and Health Survey results, POLYMOD data, and other household-level data to generate synthetic age-specific contact matrices for India. Kumar et al. [7] reported a social contact survey limited to Haryana, North India. At the time of writing, raw data from neither study was available in the form needed to generate matrices for our model. However, similarly to the POLYMOD matrix, both studies found strong assortativity for in-age-group contacts, with additional assortativity between younger adults and children. Therefore, we speculate that differential impact across update methods is likely to remain stable, despite possible different magnitudes of vaccine impact and age-specific impact patterns. Furthermore, we found that adjustment for different household structure, classroom, and workplace composition did not substantially change the assortativity patterns of our base contact matrix. This analysis suggests that while household structure in India will likely differ from POLYMOD countries, at nationally aggregated levels with the age groups used in our study, these differences are unlikely to substantially bias our results.

Finally, we calibrated the M0, M2, and M3 models to fit the baseline predicted overall TB incidence rate in 2050 of the M1 model. This likely reduced the parameter space available to calibrate the M0, M2, and M3 models. However, this deliberate constraint allowed us to isolate the effects of differential contact matrix updates on vaccine impact by maintaining comparable baseline trajectories between the four models. Thus, we assumed that relative differences in vaccine impact between M0–M3 were preserved at the cost of an error in absolute magnitudes. Further, we assumed a constant probability of infection per infectious contact in all age groups. Children are believed to be less infectious than adults or the elderly [27]. Thus, independently calibrating this parameter for each age group may magnify the effects of changes to the contact matrix; however, as the contribution to transmission from children is very small, this is unlikely to affect our findings substantially.

Our findings also likely reflect characteristics of tuberculosis' natural history. TB disease can recur through either reactivation from latency or relapse from the recovered state; thus,

some fraction of disease remains resistant to contact and transmission changes. Both latency and recovered state may persist for many years, introducing lag time between changes in transmission dynamics and changes in disease burden. It would be interesting to carry out similar experiments with other long-duration infections.

Most empirical contact studies—including POLYMOD—report rates of "close contact", i.e., a social contact involving physical touch or sustained conversation. Close contacts are likely to represent those required for transmission of direct contact or droplet infection. However, Mtb transmission is airborne; the pool of potentially infectious contacts is likely to be larger and include, for example, individuals who 'share the same air' in poorly ventilated spaces [28,29]. However, contact data for such 'casual contacts' is very sparse. Data from South Africa report that age-assortativity patterns in the *number* of contacts is similar between 'close' and 'casual' contacts, but contact *time* is more age-assortative in 'casual' contacts [29]. If this finding holds in settings outside of South Africa, then our study—which uses contact rates but not time—is less likely to be biased by using 'close contact'-based data. However, if further empirical data on 'casual contacts' or their differences with 'close contacts' emerges, this would be a useful avenue for further research.

In the Indian context modelled here, it is likely that projected demographic change combined with the age-specific pattern of TB contributed to the lack of observed difference in vaccine impact. The Indian population is projected to age, with an increased fraction of elderly compensated by a reduction in children. The adult group, responsible for most TB transmission and burden, remained relatively stable, both in absolute and fractional terms. As such, the specific changes the fraction of children and the elderly, though substantial, had a smaller impact on the TB epidemic and vaccine impact. Further work is required to investigate whether our findings are generalisable, in particular to settings with more significant changes to demographic composition, with different patterns of age-specific disease burden, and with diseases with shorter time courses. Finally, in this study, we investigated whether varying methods of updating contact matrices to match demography differentially affected vaccine impact estimates. However, for studies that aim to accurately *estimate* disease burden or intervention impact, country-specific contact structures grounded in data will likely be important to accurately characterise age-specific mixing.

## Conclusions

We found that model-based TB vaccine impact estimates were relatively insensitive to demography-matched contact matrix updates in an India-like demographic and epidemiologic scenario. Current model-based TB vaccine impact estimates may be reasonably robust to the lack of contact matrix updates, but further research is needed to confirm and generalise this finding. Further work is also required to investigate whether this result can be generalised to other epidemiologic and demographic contexts and other diseases.

## Supporting information

**S1 Text. Model description, specification, calibration, and supplementary results.** Fig A. Model diagram. Fig B. Contact matrix update analysis. Fig C. MCMC chains—M0. Fig D. MCMC chains—M1. Fig E. MCMC chains—M2. Fig F. MCMC chains—M3. Fig G. Posterior distributions of model parameters. Fig H. Calibration and Baseline Projections—M1. Fig I. Calibration and Baseline Projections—M0. Fig J. Calibration and Baseline Projections—M2. Fig K. Calibration and Baseline Projections—M3. Fig L. Model demography. Fig M. Disaggregated TB Incidence. Fig N. Comparison: Prem et al vs unadjusted POLYMOD. Table A.

Model Parameters. Table B. Calibration Targets. Table C. Vaccine impact.
(PDF)

## Author Contributions

**Conceptualization:** Chathika Krishan Weerasuriya, Rebecca Claire Harris, Christopher Finn McQuaid, Gabriela B. Gomez, Richard G. White.

**Formal analysis:** Chathika Krishan Weerasuriya, Richard G. White.

**Investigation:** Chathika Krishan Weerasuriya.

**Methodology:** Chathika Krishan Weerasuriya, Christopher Finn McQuaid, Richard G. White.

**Project administration:** Chathika Krishan Weerasuriya.

**Resources:** Chathika Krishan Weerasuriya.

**Software:** Chathika Krishan Weerasuriya.

**Supervision:** Rebecca Claire Harris, Christopher Finn McQuaid, Gabriela B. Gomez, Richard G. White.

**Visualization:** Chathika Krishan Weerasuriya.

**Writing – original draft:** Chathika Krishan Weerasuriya.

**Writing – review & editing:** Chathika Krishan Weerasuriya, Rebecca Claire Harris, Christopher Finn McQuaid, Gabriela B. Gomez, Richard G. White.

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
