## [Decision Letter · Decision Letter 0]

7 Dec 2021

Dear Dr Weerasuriya,

Thank you very much for submitting your manuscript "Updating contact structures to match evolving demography in a dynamic mathematical model of tuberculosis vaccination" for consideration at PLOS Computational Biology.

As with all papers reviewed by the journal, your manuscript was reviewed by members of the editorial board and by several independent reviewers. In light of the reviews (below this email), we would like to invite the resubmission of a significantly-revised version that takes into account the reviewers' comments.

Reviewer 2 raises important points regarding the reasoning and motivations that led to the choices in translating empirical data on contact matrices (main suggestion 1). We believe the issues raised require detailed attention from the authors since they represent major weaknesses in this manuscript.

We cannot make any decision about publication until we have seen the revised manuscript and your response to the reviewers' comments. Your revised manuscript is also likely to be sent to reviewers for further evaluation.

Sincerely,

Claudio José Struchiner, M.D., Sc.D.

Associate Editor

PLOS Computational Biology

Virginia Pitzer

Deputy Editor-in-Chief

PLOS Computational Biology

Reviewer 2 raises important points regarding the reasoning and motivations that led to the choices in translating empirical data on contact matrices (main suggestion 1). We believe the issues raised require detailed attention from the authors since they represent major weaknesses in this manuscript.

Reviewer's Responses to Questions

**Comments to the Authors:**

Reviewer #1: This manuscript presents a well written if rather technical modelling analysis of the impact of assumptions of contact structures on results of a mathematical transmission model of TB in India. In essence, the authors' work shows that they don't, so it is a "negative" result.

I am not an expert in TB modelling, so I cannot assess the relevance and appropriateness of the modelling assumptions, although they appear to make sense.

One major issue is that their assessment relies quite heavily on a contact structure from the POLYMOD paper, which is based on data collected from Europe. It would be important to conduct some form of sensitivity analysis, e.g. using possibly data from a different setting, which might be a better reflection of social interactions in India. Finally it was also not clear to me whether the level of future vaccination coverage could play a role for results.

It would also be important to make the model code available to other researchers in a public depostory.

Minor comments

p. 3 l.45 Vaccines can hardly be considered to be new tools. Please rephrase.

p. 23 Fig. 1 Describe what C, A & E means in legend.

p. 27 Update Citation 8. This has in the mean time been published in PNAS

Appendix:

p. 6 penultimate line Update Figure crossref.

p. 9 Figure 2 C: Please detail which lines are overlaid.

p. 21 line 2. sentence needs to be completed.

p. 28 missing in fastprogression

p. 29 missing space in adulttargeted

Reviewer #2: PCOMPBIOL-D-21-01594

The authors use and adapt a version of a previously developed and published model to explore whether temporal changes in age-specific contact patterns will have implications for the population-level impact of adult TB vaccines. The authors model four different ways in which age-specific contacts change over time, which is adapted from a previous study exploring how empirical data on age-specific contact can be applied in general. Specifically, the authors explore models that incrementally

(i) forces "reciprocity" in the number of between-age-group contacts at the size of age-groups change;

(ii) preserves "assortativity" in between-age-group contacts or the propensity to contact someone in certain age-group over others; and

(iii) preserves the total number of contacts in the population as the population sizes change.

The methods are very well detailed in the supplementary materials, and the findings are described fairly well.

The authors find that in the context of the epidemiological impact of adult TB vaccines that is deployed in a high burden setting like India, these do not significantly change the results.

Overall, I find this study methodologically sound, and the questions addressed to be of significance (but perhaps not as well motivated).

My main suggestions for the authors are as follows:

1. The main question, why and how these specific features about translating empirical data on contact matrices matter, is not motivated adequately.

This could partially be addressed by (i) describing in what form these empirical data are, and why precisely these modeling choices have to be made -- currently, it relies on the authors reading previous papers to understand these precisely;

(ii) laying out the potential reasonings behind a specific model (beyond just to get the numbers to add up); and

(iii) potential implications of the specific modeling choices, especially in the context of vaccines.

As it stands, it feels more like an extended sensitivity analysis, attempting to cover all bases -- but if framed better,

it could get at important questions at the heart of the matter about what empirical data are telling us, and why they matter.

2. Why POLYMOD? Authors justify that India-specific data were not available in a fully representative form (though arguably a state within India is more representative that European nations), but why not data from a similarly densely populated China or has household structures that are more in line with Kenya?

The specific choice of contact matrix data may not matter for the main question, but I think it worth authors addressing this choice.

3. To what degree are diary-based contact patterns applicable to TB, where the transmission is airborne?

This is a general comment on the use of these data -- I would appreciate if the authors could provide their take on this: to what degree this is a data limitation (and these are the best data we have), or

whether they believe that these data are fairly good representation of the age-specific contact patterns.

4. Are the results generalizable? I think it would be great if the authors could speak to:

(i) whether their finding (no significant difference) is tied to the choice of contact matrix (also relates to point 2), and/or

(ii) the projected demographic changes in India.

One suggestion here, if logistically possible, would be to conduct sensitivity analyses with one of the alternative contact matrix data and or demographic changes.

Minor points.

5. Title: Suggest specifying age-specific contact structure, since contact structure can be based on other non-age related factors

6. Fig 1: Suggest spelling out C, A, and E.

7. Introduction, lines 62-65: Please include references: it will also make help make authors point more clear.

Reviewer #3: Please see the attached review letter.

**Have the authors made all data and (if applicable) computational code underlying the findings in their manuscript fully available?**

Reviewer #1: **No: **

Reviewer #2: None

Reviewer #3: Yes

PLOS authors have the option to publish the peer review history of their article (what does this mean?). If published, this will include your full peer review and any attached files.

Reviewer #1: No

Reviewer #2: No

Reviewer #3: No
---

## [Decision Letter · Decision Letter 1]

28 Feb 2022

Dear Dr Weerasuriya,

Thank you very much for submitting your manuscript "Updating contact structures to match evolving demography in a dynamic mathematical model of tuberculosis vaccination" for consideration at PLOS Computational Biology. As with all papers reviewed by the journal, your manuscript was reviewed by members of the editorial board and by several independent reviewers. The reviewers appreciated the attention to an important topic. Based on the reviews, we are likely to accept this manuscript for publication, providing that you modify the manuscript according to the review recommendations.

Reviewer 2 has a few very minor suggestions for your consideration prior to acceptance.

Sincerely,

Claudio José Struchiner, M.D., Sc.D.

Associate Editor

PLOS Computational Biology

Virginia Pitzer

Deputy Editor-in-Chief

PLOS Computational Biology

[LINK]

Reviewer's Responses to Questions

**Comments to the Authors:**

Reviewer #1: All comments have been adressed.

Reviewer #2: Authors have revised the manuscript to meaningfully address or provide thoughtful additions to all reviewer comments.

Minor points.

1. Suggest including "age-specific" in the title.

2. line 94, "interact with" is written twice

3. Suggest including a caveat that although these specific results were not affected, including county-specific contact patterns could be important for general transmission modeling with age-specific contact patterns, just as incorporating evolving demographic changes.

Reviewer #3: Thank you for addressing the comments raised in my initial review.

**Have the authors made all data and (if applicable) computational code underlying the findings in their manuscript fully available?**

Reviewer #1: Yes

Reviewer #2: None

Reviewer #3: Yes

PLOS authors have the option to publish the peer review history of their article (what does this mean?). If published, this will include your full peer review and any attached files.

Reviewer #1: No

Reviewer #2: No

Reviewer #3: No

Figure Files:

Data Requirements:

Reproducibility:

References:

---

## [Editor Report · Decision Letter 2]

8 Mar 2022

Dear Dr Weerasuriya,

We are pleased to inform you that your manuscript 'Updating age-specific contact structures to match evolving demography in a dynamic mathematical model of tuberculosis vaccination' has been provisionally accepted for publication in PLOS Computational Biology.

Best regards,

Claudio José Struchiner, M.D., Sc.D.

Associate Editor

PLOS Computational Biology

Virginia Pitzer

Deputy Editor-in-Chief

PLOS Computational Biology

---

## [Editor Report · Acceptance letter]

19 Apr 2022

PCOMPBIOL-D-21-01594R2 

Updating age-specific contact structures to match evolving demography in a dynamic mathematical model of tuberculosis vaccination

Dear Dr Weerasuriya,

I am pleased to inform you that your manuscript has been formally accepted for publication in PLOS Computational Biology. Your manuscript is now with our production department and you will be notified of the publication date in due course.

With kind regards,

Agnes Pap
